# Brazilian Clinical Strains of *Actinobacillus pleuropneumoniae* and *Pasteurella multocida*: Capsular Diversity, Antimicrobial Susceptibility (*In Vitro*) and Proof of Concept for Prevention of Natural Colonization by Multi-Doses Protocol of Tildipirosin

**DOI:** 10.3390/antibiotics12121658

**Published:** 2023-11-25

**Authors:** Suzana Satomi Kuchiishi, Simone Ramos Prigol, Eduarda Bresolin, Bianca Fernandes Lenhard, Caroline Pissetti, María-José García-Iglesias, César-Bernardo Gutiérrez-Martín, Sonia Martínez-Martínez, Luiz Carlos Kreutz, Rafael Frandoloso

**Affiliations:** 1Laboratory of Microbiology and Advanced Immunology, Faculty of Agronomy and Veterinary Medicine, University of Passo Fundo, Passo Fundo 99052-900, Brazil; suzana@cedisa.org.br (S.S.K.); dudabresolin98@gmail.com (E.B.); biancaflenhard@gmail.com (B.F.L.); lckreutz@upf.br (L.C.K.); 2Centro de Diagnóstico de Sanidade Animal—CEDISA, Concórdia 89727-000, Brazil; carolpissetti@gmail.com; 3AFK Imunotech, Passo Fundo 99052-900, Brazil; simoneramosprigol@gmail.com; 4Animal Health Department, Faculty of Veterinary Medicine, University of León, 24007 León, Spain; mjgari@unileon.es (M.-J.G.-I.); cbgutm@unileon.es (C.-B.G.-M.); smarm@unileon.es (S.M.-M.)

**Keywords:** *Actinobacillus pleuropneumoniae*, *Pasteurella multocida*, *Glaesserella parasuis*, typing, virulence genes, antimicrobial susceptibility pattern, tildipirosin, colonization, pig

## Abstract

One hundred *Actinobacillus pleuropneumoniae* (App) and sixty *Pasteurella multocida* subsp. *multocida* serogroup A (PmA) isolates were recovered from porcine pneumonic lungs collected from eight central or southern states of Brazil between 2014 and 2018 (App) or between 2017 and 2021 (PmA). *A. pleuropneumoniae* clinical isolates were typed by multiplex PCR and the most prevalent serovars were 8, 7 and 5 (43, 25% and 18%, respectively). In addition, three virulence genes were assessed in *P. multocida* isolates, all being positive to *capA* (PmA) and *kmt1* genes, all negative to *capD* and *toxA*, and most of them (85%) negative to *pfhA* gene. The susceptibility of both pathogens to tildipirosin was investigated using a broth microdilution assay. The percentage of isolates susceptible to tildipirosin was 95% for App and 73.3% for PmA. The MIC_50_ values were 0.25 and 1 μg/mL and the MIC_90_ values were 4 and >64 μg/mL for App and PmA, respectively. Finally, a multiple-dose protocol of tildipirosin was tested in suckling piglets on a farm endemic for both pathogens. Tildipirosin was able to prevent the natural colonization of the tonsils by App and PmA and significantly (*p* < 0.0001) reduced the burden of *Glaesserella parasuis* in this tissue. In summary, our results demonstrate that: (i) tildipirosin can be included in the list of antibiotics to control outbreaks of lung disease caused by App regardless of the capsular type, and (ii) in the case of clinical strains of App and PmA that are sensitive to tildipirosin based on susceptibility testing, the use of this antibiotic in eradication programs for *A. pleuropneumoniae* and *P. multocida* can be strongly recommended.

## 1. Introduction

Porcine Respiratory Disease Complex (PRDC) is a worldwide distributed multifactorial disease with a major impact on swine production. The acute form of PRDC is associated with high mortality whereas the chronic pleuropneumonia form reduces weigh gain and its control usually requires the use of medication resulting in severe economic losses during swine husbandry [1,2]. Procedures for decreasing the incidence of PRDC are based on correct biosecurity protocols, good management practices, vaccination, and a proper antimicrobial treatment to limit the severity of its clinical form [2]. The pathogens most frequently found associated with PRDC are *Actinobacillus pleuropneumoniae*, *A. suis*, *Pasteurella multocida*, *Bordetella bronchiseptica*, *Streptococcus suis*, *Glaesserella parasuis* and *Mycoplasma hyopneumoniae* [3].

*A. pleuropneumoniae*, a member of the *Pasteurellaceae* family, is considered a primary pathogen and usually affects finishing pigs. Based on the requirement of nicotinamide adenine dinucleotide (NAD) for *in vitro* growth, *A. pleuropneumoniae* can be differentiated into two biovars; however, using multiplex PCR designed for capsular typing, it can be classified into 19 different serovars [4] with distinct geographical distribution and marked differences in their ability to cause disease. *Pasteurella multocida* is one of the most common bacterial agents isolated from respiratory clinical diseases in pig husbandry [5], but it behaves as a secondary pathogen agent in PRDC [6]. Some clinical strains of *P. multocida* serogroup A have been also described as a primary agent of bronchopneumonia, serositis and hemorrhagic septicemia in pigs [7], and the prevalence of *P. multocida* serogroups varies from region to region and over time within a certain site [8]. Additionally, in combination with *B. bronchiseptica*, *P. multocida* can cause progressive atrophic rhinitis, a disease that can be prevented by using licensed vaccines. Toxin producing strains of *P. multocida* associated with atrophic rhinitis are most usually from serogroup D and less frequently from serogroup A. The serogroups that produce pneumonia are routinely non-toxigenic and their proportion is usually higher than the toxigenic strains [9]. Outbreaks of lung diseases caused by *P. multocida* during the growing-finishing phase are quite frequent in Brazil [10] forcing veterinarians to use different antimicrobial molecules to control the infection.

The prophylactic and metaphylactic use of antimicrobials during pig husbandry is not yet prohibited or even controlled in Brazil. However, the use of antimicrobial molecules such as lincomycin, tiamulin, and tylosin as growth promoters have been prohibited in Brazil in 2020 [11] and this represents the first step towards reducing the use of antibiotics in the pork production system. Nonetheless, the use of vaccines against respiratory, systemic, and enteric pathogens is also widely spread in the Brazilian production system. However, due to the characteristics of our production system, which includes comingling piglets from many different origins, the use of antimicrobials as a tool to control respiratory infections is a common practice in the majority of Brazilian pig farms use. Although important to control infections, the widespread and inappropriate use of antimicrobials might lead to the surge of resistance traits within different bacteria species and compromise the therapeutic use of specific antimicrobials.

The prudent use of antibiotics in animal production and human medicine is a priority objective of the World Health Organization [12]. Notably, the pharmaceutical industry has not introduced any new classes of antibiotics in the past three decades, posing a significant challenge in combating invasive multi-resistant bacteria. Some pharmaceutical companies have made molecular modifications to existing active ingredients, enhancing their potency. This is the case of tildipirosin (20,23-dipiperidinyl-mycaminosyl-tylonolide), a semi-synthetic tylosin analog that inhibits the function of the 23S ribosomal ribonucleic acid (rRNA) of the 50S ribosomal subunit of bacterial cells [13]. Tildipirosin was developed by MSD Animal Health and introduced in international markets under the name Zuprevo in 2011 [14]. This antibiotic, available only as injectable presentation, has high *in vitro* efficacy against *A. pleuropneumoniae*, *G. parasuis* and *Pasteurella multocida* [15,16,17], and represents an excellent weapon for the treatment of these important infectious agents.

In this scenario and considering the importance of evaluating the susceptibility profile of disease-causing bacteria to injectable antibiotics in pig husbandry, in this work we address two objectives. Firstly, we conducted the molecular typing and virulence characterization of a broad panel of Brazilian clinical strains of *A. pleuropneumoniae* and *P. multocida*. Subsequently, we assessed the *in vitro* efficacy of tildipirosin against these two pathogens and finally, we evaluated the ability of tildipirosin to prevent tonsillar colonization in piglets reared in a farm with endemic circulation of *A. pleuropneumoniae*, *G. parasuis* and *P. multocida*.

## 2. Results

### 2.1. Typing of Actinobacillus pleuropneumoniae Clinical Isolates

In Brazil, there is a scarcity of scientific data regarding the distribution of *A. pleuropneumoniae* serotypes associated with clinical outbreaks of porcine pleuropneumonia or lung lesions observed during the inspection of carcasses at the slaughterhouse. This lack of information has motivated us to conduct this analysis. Based on the molecular capsule-specific typing, we found that the majority of *A. pleuropneumoniae* clinical isolates were classified as serovar 8 (43%), followed by serovar 7 (25%) and by serovar 5 (18%). Altogether, these three serovars represented 86% of the clinical strains *of A. pleuropneumoniae* assessed. The remaining serovars detected (SV1, SV6, SV14) represented each less than 5% of the total typeable isolates. Finally, 5 clinical strains could not be typed by the multiplex PCRs performed (Figure 1).

As illustrated in Figure 2, serovar 8 spread across the eight Brazilian states studied, and clearly represents the most frequently isolated serovar throughout the study. A total of 56.7% of all strains from Rio Grande do Sul (RS) belonged to this serovar, while its frequency was 28.6% in Santa Catarina (SC), 45.5% in Paraná (PR) and 41.7% in Minas Gerais (MG). The states of RS, PR and MG showed a greater diversity of *A. pleuropneumoniae* serovars compared to the other five states studied. In these states it was possible to detect SVs 1, 5, 7, 8, 14 and NT. In MG, SV14 was not found, however, SV6 was exclusively detected. In the central-west region of Brazil, comprising Mato Grosso (MT), Mato Grosso do Sul (MS) and Goiás (GO) states, the presence of serovars 7 and 8 was detected in MT and MS. In contrast, in GO only the serovar 8 was detected, which demonstrates, overall, a low diversity of serovars circulation in this region, which can be justified by its geographical location (away from the southern region that has the highest density of pigs) and the lower flow of animals from other states. Similarly, low diversity was observed in São Paulo (SP), where only SV 8 and 7 were detected. Curiously, in SC, a state located between RS and PR, only serovars 5, 7 and 8 were detected, representing a lower diversity compared to its neighboring states. Among these serovars, 46.4% of the collected isolates belonged to serovar 5 (Figure 2).

Among the serovars originating from farms, most belonged to serovar 8 (44.20% compared to 42.10% from slaughterhouses) or to serovar 14 (9.30% compared to 1.80% from slaughterhouses). On the other hand, for serovars 5 (21.10% from slaughterhouses compared to 14.00% from farms) and 7 (26.30% from slaughterhouses compared to 23.25% from farms) the opposite was observed. In addition, of the majority of serovar 5 isolates were observed in 2016–2017 (66.70%), whereas those of serovar 8 were predominant during 2015–2016 (55.80%).

### 2.2. Determination of Virulence-Associated Genes in Pasteurella multocida Isolates

*Pasteurella multocida* is a respiratory bacterium that has been recognized as one of the main pathogens associated with dry pleurisy lesions in Brazil; leading to economic losses that extend beyond the impact on animal performance at the farm. Among the two capsular genes examined, *capA* was harbored by 100% of isolates, whereas *capD* was not detected in any of the isolates. In contrast, the *kmt1* gene, responsible for encoding an esterase/lipase, was detected in all the 60 (100%) isolates. The gene *toxA* (dermonecrotoxin A) was not found in any of the isolates, and the *pfhA* gene, which codes for a filamentous hemagglutinin, was present in only 16,7% (*n* = 10) of the *P. multocida* clinical strains. Of these, 50% of the *pfhA*^+^ strains were from the state of Rio Grande do Sul. The other *pfhA*^+^ strains were detected in the states of Santa Catarina (*n* = 2, 20%), Paraná (*n* = 2, 20%) and São Paulo (*n* = 1, 10%) (Figure 3).

### 2.3. Antimicrobial Susceptibility Testing in Actinobacillus pleuropneumoniae and Pasteurella multocida Isolates to Tildipirosin

The sensitivity profile of *A. pleuropneumoniae* and *P. multocida* serogroup A can be evaluated using an antibiogram and/or minimum inhibitory concentration (MIC) tests. The second test allows evaluating the evolution of the sensitivity profile of the microorganism over time or for comparing the profile between different microorganisms of the same species at a given time. In our study, we used the MIC testing, and the quality control values were within the CLSI quality control range for tildipirosin. The range, MIC_50_, MIC_90_ and antimicrobial susceptibility of the 100 isolates recovered from Brazil from 2014 to 2018 are shown in Table 1. The susceptible breakpoint for tildipirosin, when testing *A. pleuropneumoniae* isolates, is 16 μg/mL, and we found that only 5% of isolates were not susceptible to this macrolide. In Table 1, we also provide the cumulative percentage of isolates inhibited by a given drug concentration plus all concentrations lower than it. In this context, the growth of at least 50% of the bacterial isolates was hampered by ≤0.25 μg/mL tildipirosin, and 91% of them were inhibited by 4 μg/mL. Therefore, MIC_50_ and MIC_90_ were 0.25 and 4 μg/mL, respectively. The MIC ranged from ≤0.06 to >64 μg/mL. In addition, the concentration that resulted in the highest inhibition of *A. pleuropneumoniae* isolates was 0.25 μg/mL, with 32 clinical strains affected, followed by 0.5 μg/mL, affecting 27 *A. pleuropneumoniae* strains. About 44.4% of serovar 5 isolates were precisely inhibited by this latter tildipirosin concentration, and more than a third of serovar 5 and serovar 8 were inhibited by 0.25 μg/mL. The five resistant *A. pleuropneumoniae* isolates belonged to serovar 1, 7 and 8.

The range, MIC_50_, MIC_90_ and antimicrobial susceptibility of the 60 *P. multocida* isolates collected in Brazil from 2017 to 2021 are also provided in Table 1. Tildipirosin was less effective against *P. multocida* compared to *A. pleuropneumoniae* clinical isolates; the resistance rate of *P. multocida* was higher and was detected in 16 (26.7%) isolates. At least 11 (68.7%) of the resistant isolates were recovered from growing-finishing pigs. The data related to the MIC parameters confirm the lower susceptibility rate of *P. multocida*, since the MIC_50_ was four times higher (1 μg/mL) compared to the MIC_50_ found for *A. pleuropneumoniae,* and the MIC_90_ was 16 times higher (64 μg/mL) than that found for *A. pleuropneumoniae.* The MIC range varied from 0.5 to 64 μg/mL. The concentration that clearly inhibited the growth of the largest number of *P. multocida* isolates was 1 μg/mL, affecting 28 clinical strains (46.7% of the total). Even so, 23 other *P. multocida* isolates were effectively inhibited by concentrations higher than 1 μg/mL. Highly resistant strains to tildipirosin (>64 μg/mL) were detected in the western region of Santa Catarina (*n* = 5), in the northern and northwest (*n* = 1) regions of Rio Grande do Sul (*n* = 5 and 1, respectively), in the western region of Paraná (*n* = 2), in the northwest region of São Paulo (*n* = 1) and in the central-north region of Mato Grosso (*n* = 1).

### 2.4. In Silico Association Analysis

No significant differences were observed when comparing the results obtained for the *A. pleuropneumoniae* antimicrobial susceptibility testing and the distribution of clinical isolates regarding Brazilian states, the year of isolation, origin (from farms or slaughterhouses), or serotype. Similarly, no significant differences were observed either between the *P. multocida* tildipirosin susceptibility testing and each of the following variables: Brazilian states, year of isolation, pig age range, or the presence or absence of the *pfhA* gene (Appendix A).

### 2.5. In Vivo Efficacy of Tildipirosin

Due to the high susceptibility rates of *A. pleuropneumoniae* and *P. multocida* strains to tildipirosin (Table 1), we decided to evaluate the effectiveness of this antibiotic in controlling bacterial colonization of tonsils in piglets reared on a farm with endemic circulation of *A. pleuropneumoniae*, *P. multocida* and *G. parasuis*. As illustrated in Figure 4, antibiotic treatment administered to sows and piglets prevented or eliminated the colonization of *A. pleuropneumoniae* and *P. multocida* in 100% of treated animals. In contrast, 35% and 60% of untreated piglets (from untreated sows) were colonized by *A. pleuropneumoniae* and *P. multocida* at 21 days of age, respectively. The same trend was observed with *G. parasuis*, however, all sampled tonsils (100%) of untreated piglets were colonized by *G. parasuis,* and the treatment reduced the frequency of positive animals by 60% compared to untreated animals. 

As illustrated in Figure 5, the load of *G. parasuis* found in positive samples from treated animals was significantly lower (*p* < 0.0001) compared to untreated animals. Regarding the dynamics of infection, we observed that it was progressive over time for *A. pleuropneumoniae* and *P. multocida* in untreated animals. At 7 days of age, we already detected piglets with both microorganisms in the tonsils surface (Figure 4). The dynamics of *G. parasuis* infection was different and more intense; in this case, 100% of the piglets were already colonized at 7 days of age, and the load of *G. parasuis* associated with the tonsil mucosa, expressed as a PCR cycle number (Ct), increased over the first 3 weeks of life (Figure 5).

## 3. Discussion

A collection of 100 *A. pleuropneumoniae* isolates recovered from Central and Southern Brazil was subjected to molecular typing and susceptibility testing for tildipirosin. The prevalence of serovars 8 and 7 was most pronounced and should be highlighted. The prevalence situation of *A. pleuropneumoniae* in Brazil partly reflects the trend seen in Europe over the last two decades. For instance, in 2009, serovar 8 was by far the most frequently isolated serovar (78.0%) [18], and in England and Wales in 2008–2014, it accounted for 71.7% of isolates [19]. In Hungary, before and between 2012 and 2016, the prevalence of serovar 2 (43% and 38.9%) was followed by serovar 13 (43% and 10.4%) and serovar 8 (0% and 10.4%) [20]. In Quebec, a report from 2020 showed that serovar 7 was the most prevalent [21], but in an earlier report from 2015, serovar 5 ranked first, and serovar 7 ranked second among all isolates [22]. The situation in Asia differed from that of Brazil, as the predominant *A. pleuropneumoniae* isolates belonged to serovars 1, 2 or 5 with a minimal percentage of serovar 7 strains [23,24]. We emphasize that the detection of SV14 had never been observed in Brazil before, which indicates that its introduction may, hypothetically, have occurred mainly due to: (i) the introduction of animals from other countries (genetic improvement); and (ii) wild boar movement, as demonstrated previously [25].

The fact that 5% of isolates were non-typeable in our investigation indicates that the causal agent of pleuropneumonia could be differentiated into a higher number of serovars higher than that reported so far. Anyway, the diversity of *A. pleuropneumoniae* serovars shown here is similar to that described in most reports [18,19,20,22]. However, this variety was lower (only four serovars) in another study [24], and considerably higher (up to ten serovars) in other recently published studies [21].

*A. pleuropneumoniae* isolates recovered from pigs that died or were sacrificed due to severe respiratory distress on the farms are usually associated with acute cases of pleuropneumonia since they did not reach slaughterhouse. Conversely, those recovered from pigs with pneumonic lesions at abattoir are associated with a chronic infection form. However, quite the opposite, the highest percentage of serovar 8 isolates (that expresses both ApxII and ApxIII toxins) [26] which is considered as a not too virulent serovar, was detected in diseased animals at the farms level. Meanwhile, the greatest amount of serovar 5 isolates were recovered from pigs slaughtered at the abattoir. This finding suggests the implication of serovar 5 in the chronic forms of the disease. Although, due to its high virulence (by secretion of ApxI and ApxII toxins [27]), we cannot rule out that some animals did not develop acute forms of the disease during growing-finishing phase. Detailed information on all batches of animals from which the isolated strains came was not available.

In addition, a total of 60 *P. multocida* isolates from the same Brazilian geographical locations were studied for some virulence factors and susceptibility to tildipirosin. As expected, all were classified as belonging to capsular type A mainly because *P. multocida* were recovered from cases of pneumonia and not from atrophic rhinitis, consistent with previous studies [9,28,29]. Thus, as expected, the gene encoding dermonecrotoxin A (*toxA* gene) was absent in all isolates, because this gene and the toxin codified by it have been mostly detected in *P. multocida* capsular type D strains [29]. Nonetheless, exceptions to this have been described; for instance, *toxA* gene was also detected in 12% of *P. multocida* capsular type A isolates in an investigation carried out in Spain from 2017 to 2019 by part of our investigation team (unpublished results). The presence of *pfh*A gene was observed in the current study in ten *P. multocida* isolates (10%), a rate similar to that reported previously by other investigators [7,9,30]. However, this hemagglutinin-encoding gene was found in 95.3% of the *P. multocida* recovered in India [31]. In a Brazilian study [32], this gene was detected in all five highly pathogenic strains of *P. multocida* as primary pathogen causing disease in pigs without any co-infection.

Brazil is known worldwide as a major pig producer, ranking as the fourth-largest producer and exporter. The country has a swine population of over two million sows [33]. Out of 30 countries around the world (26 in Europe, 2 in Oceania, and 2 in America), Brazil is the second-highest consumer of antibiotics in pig farming, with a rate of 358.4 mg/kg of biomass, trailing only Cyprus [11,34]. The continued overuse of antibiotics, including for prophylactic and metaphylactic purposes, and their unreasonable dosages have raised concerns. This overuse can lead to the selection of antimicrobial-resistant bacteria, which can spread not only among animals but also to humans [35]. Effective antimicrobial agent selection is crucial for pig production, especially against respiratory diseases, as those caused by the bacteria belonging to the Porcine Respiratory Disease Complex (PRDC).

Tildipirosin, a semisynthetic derivative of tylosin, is a latest generation 16-membered ring macrolide that is used to combat infectious diseases caused by respiratory pathogens, including *A. pleuropneumoniae, G. parasuis* and *P. multocida*, all being part of the PRDC [35]. Tildipirosin works by binding to the 23S ribosomal RNA, which is part to the 50S subunit of bacterial ribosome, thus inhibiting bacterial protein synthesis [36]. This antibiotic exhibit long-lasting, potent bacteriostatic action, high bioavailability, and achieves high drug concentrations in lung tissue, making it particularly effective [17]. Our study showed that tildipirosin was more effective against *A. pleuropneumoniae* compared to *P. multocida* organisms, with 5% *versus* 26.7% non-susceptible isolates, respectively.

In a Spanish study from 2020 that assessed 162 *A. pleuropneumoniae* strains isolated from respiratory clinical cases, the rate of susceptibility rate to tildipirosin was 99.4% [16], approximately four points higher than our findings. However, the Spanish study indicated somewhat higher MICs, MIC_50_ and MIC_90_ values compared to Brazilian isolates, suggesting that higher tildipirosin concentrations may be required in Spain to effectively inhibit *A. pleuropneumoniae* growth.

Regarding *P. multocida*, our investigation revealed a higher non-susceptible rate (26.7%) compared to the 2.3% among 130 *P. multocida* isolates collected in Spain from 2017 to 2019 [16]. Despite the much greater MIC_90_ value for Brazilian *P. multocida* strains, the MIC range and MIC_50_ matched in both studies [16]. Nevertheless, another study involving *P. multocida* isolates from China showed considerably lower MIC values for these parameters compared to our findings [17]. No resistance to tildipirosin was detected in 48 *P. multocida* isolates from North-western Spain in the 2017–2019 period, where 0.5 μg tildipirosin/mL inhibited most of them (43.7%) [16]. In contrast, our study found that 46.7% of Brazilian *P. multocida* clinical isolates required twice concentration to inhibit their growth.

The existence of *P. multocida* isolates with MICs over 16 μg/mL, observed in 16 clinical isolates in the current study, has previously been related with the presence of macrolide resistance genes, such as *erm(42)* or *msr(E) and mph(E)* [17]. These resistance genes have been reported for tildipirosin and other macrolides commonly used in veterinary medicine by Michael et al. [37] and Poehlsgaard et al. [36]. The presence of resistant isolates is becoming an increasingly serious problem due to the rapid spread to common antimicrobial compounds [38]. While cross-resistance between macrolides would be a logical explanation for understanding the emergence of resistant isolates to tildipirosin in Brazil, further studies should be conducted to elucidate the evolutionary process of *A. pleuropneumoniae* or *P. multocida* resistances in regions with high resistance to lincosamides.

Due to the high efficacy of tildipirosin, especially against *A. pleuropneumoniae* (Table 1) and *G. parasuis* [15], we decided to evaluate the capacity of this antibiotic to eliminate the colonization of bacteria related to PRDC and that belong to the Pasteurellaceae family. *A. pleuropneumoniae* and *P. multocida* serogroup A are two important microorganisms responsible for lung disease in pigs during the growing-finishing phase, resulting in substantial economic losses related to clinical outbreaks, as well as for condemnations during the inspection of carcasses at slaughterhouses [1,10]. On the other hand, *G. parasuis* causes an acute-severe systemic disease in younger animals, especially during the nursery phase [39].

As illustrated in Figure 4, the transmission of these three microorganisms occurs during the lactation phase, with sows playing a key role in the epidemiological process of infection. Considering this, a logical strategy to mitigate transmission would be to reduce the bacterial load present in the oral and respiratory mucosa of the sows, and for that, it would be necessary to use an antibiotic that reaches high concentrations in these tissues, especially in the tonsil; and tildipirosin is a macrolide that has this characteristic [40,41]. In addition, considering that the ½ life of tildipirosin is 7 days in pigs [40], in our study we evaluated a protocol based on the application of a single dose of tildipirosin in sows, and another 3 doses of this antibiotic in suckling piglets. The first piglet’s dose being applied over the first 24 h after birth, and the others at intervals of 7 days (Figure 6), ensuring that the piglets had a maximum concentration of the antibiotic circulating throughout lactation phase.

Using this protocol, we observed that none of the treated piglets was colonized by *A. pleuropneumoniae* and *P. multocida* throughout the lactation phase, making them considered negative for these microorganisms at weaning. This observation allows us to suggest, even if in a very preliminary way, that the treatment based on the application of multiple doses of tildipirosin can be a strategic and effective tool in programs aimed to eradicate for *A. pleuropneumoniae* and *P. multocida*. Additional studies need to be performed to assess the repeatability of the results obtained in this study.

In parallel, we observed that at weaning, 40% of the piglets in the treated group were positive for *G. parasuis* colonization in the tonsils. However, the bacterial load was significantly lower (*p* < 0.0001) compared to the untreated group where 100% carried this pathogen (Figure 5). Therefore, treatment with tildipirosin reduced the number of colonized animals and the infection pressure of the positive ones. In practical terms, it will represent a lower transmission of *G. parasuis* during the first weeks of nursery phase. This reduction in *G. parasuis* colonization can help mitigate the susceptibility window of piglets to *G. parasuis*, especially in the transition from maternally derived immunity to active inductive immunity, which occurs due to vaccine administration at 21 and 35 days of age. We must also emphasize that tildipirosin was not able to prevent colonization or even reduce the load of *Streptococcus suis* present in the tonsils of the animals.

We need to highlight that the use of tildipirosin as a tool to eradicate *A. pleuropneumoniae* and *P. multocida* must be carried out with caution and aiming at improving the health status of the herd. From the point of view of reducing the amount of antibiotics in food-production animals, we understand that by eliminating these two pathogens from the respiratory tract through the strategic use of 38 mg of tildipirosin per piglet, it is possible to avoid clinical outbreaks of respiratory disease caused by these pathogens, which consequently prevents the use of higher amounts of antibiotics needed for treatment of older pigs.

As documented a few years ago, a clear association was found between interventions that restrict antibiotic use and reduction in the prevalence of antibiotic-resistant bacteria in animals and in different human subgroups. Overall, reducing antibiotic use led to a 15% reduction in the prevalence of antibiotic-resistant bacteria in animals and a 24–32% reduction in multidrug-resistant bacteria [12]. In this context, it is important to emphasize that future studies are needed to determine whether the optimized use of tildipirosin (at therapeutic dose) in suckling-piglets can or not select resistant bacterial strains against this and other macrolides, and conditionate the maintenance of this protocol whenever it does not represent a risk for animal and human health.

Since this protocol can be applied to commercial pig farms, the cost of the product is a significant consideration that can either enable or hinder its use. Considering the market price of tildipirosin in Brazil (US$ 0.24/mL), treating a 200 kg sow would cost US$ 4.8 and the 3-dose program for a piglet would cost US$ 0.21. The cost of treatment may justify its use in farms facing substantial challenges from *Pastereullaceae* family bacteria or those aiming to eradicate these microorganisms from their production system.

Finally, although most of clinical strains of *A. pleuropneumoniae* (95%) and *P. multocida* (73.3%) were susceptible to tildipirosin at *in vitro* analysis, the use of this drug to treat disease outbreaks, or to eliminate or reduce bacterial colonization of tonsils is recommended only after analyzing the sensitivity profile of the microorganisms of interest isolated from carrier animals (sows) or pigs displaying clinical signs of disease and coming from the same production flow.

## 4. Material and Methods

### 4.1. Actinobacillus pleuropneumoniae Isolates and Molecular Capsule Typing

A set of 100 clinical isolates of *A. pleuropneumoniae* was obtained from eight Central or Southern states of Brazil (Rio Grande do Sul—RS, 30%; Santa Catarina—SC, 28%; Minas Gerais—MG, 12%; Paraná—PR, 11%; São Paulo—SC, 8%; Mato Grosso—MG, 6%; Goiás—GO, 2%; and Mato Grosso do Sul—MS, 3%). These isolates were collected over the years 2014 (5%), 2015 and 2016 (26% each), 2017 (17%) and 2018 (26%). Among these, 57% were recovered from lungs at slaughterhouses, while the remaining 43% from lungs obtained from necropsies at farms. All clinical strains were recovered from pneumonic lungs (necrohemorrhagic lesions located in the diaphragmatic lobes) that were taken from growing pigs between 75 and 180 days old. The same lung pattern was selected at the slaughterhouse for sample collection. *A. pleuropneumoniae* was isolated on chocolate blood agar plates (Oxoid Ltd.a, São Paulo, Brazil) after 24 h at 37 °C under microaerophilic conditions. The identification of presumptive isolates was carried out by using standard phenotypic procedures and confirmed by multiplex PCRs for capsule typing of serovars 1 to 19 [4].

### 4.2. Pasteurella multocida Isolates, Capsular Typing, Molecular Confirmation, and Detection of Virulence-Associated Genes

A collection of 60 clinical isolates of *P. multocida* was obtained from six Central or Southern states of Brazil (Santa Catarina, 38.3%; Rio Grande do Sul, 33.3%; Paraná, 21.7%; São Paulo, 3.3%; Espírito Santo and Mato Grosso do Sul, 1.7% each). These isolates were collected over the years 2017 (16.7%), 2018 (18.3%), 2019 (45.0%), 2020 (13.3%) and 2021 (6.7%) and were included in this study. A total of 53.3% clinical strains were recovered from lungs of necropsied pigs at farms and the remaining 46.7% strains were obtained from pig’s lungs at slaughterhouses. Of the total of clinical strains, 78.3% were isolated from growing-finishing pigs (71–250 days old), 16.7% from weaned pigs (31–70 days old), and the remaining 5.0% from piglets (until 30 days old). Swabs from suspicious samples (lungs exhibiting bronchopneumonia or pneumonia) were seeded on Columbia blood agar plates (Oxoid Ltd.a, São Paulo, Brazil) containing 5% defibrinated sheep blood and incubated for 24 h at 37 °C under aerobic conditions. Bright, whitish, mucous, dewdrop-like colonies were further confirmed and typed by a multiplex capsular PCR typing system [42]. PCR was also used to investigate the presence of dermonecrotoxin *toxA* gene [43] and adhesion-related *pfhA* gene (coding for a filamentous hemagglutinin) [9].

### 4.3. Antimicrobial Sensitivity Testing

The antimicrobial susceptibility to tildipirosin was determined using the broth microdilution method in accordance with the recommendations by the Clinical and Laboratory Standards Institute [44,45], with slight modification. Colonies from each *A. pleuropneumoniae* or *P. multocida* clinical isolate were picked and homogenized in demineralized water, quantified by flow cytometry [15], and diluted in Veterinary Fastidious Medium (for *A. pleuropneumoniae)* or cation-adjusted Mueller-Hinton broth (in the case of *P. multocida*) to reach an inoculum concentration of 5 × 10^5^ bacteria/mL. Different Tildipirosin concentrations (ranging from 0.06 to 64 μg/mL) in 96-well plates were incubated with 100 μL per well of the bacteria inoculum at 37 °C for 18–20 h, under a 5–10% CO_2_ atmosphere for *A. pleuropneumoniae*, only. Each microplate was read manually, and the minimal inhibition concentration (MIC) was established as the lowest tildipirosin concentration being able to inhibit visible growth. An *A. pleuropneumoniae* quality control strain (ATCC 27090) was also included in each susceptibility assay. MIC_50_ and MIC_90_ were defined as MICs inhibiting 50% or 90% of the isolates, respectively. The clinical breakpoints for tildipirosin were obtained from the literature [16]: 16 μg/mL for *A. pleuropneumoniae* and 4 μg/mL for *P. multocida*, and these bacterial isolates were considered as susceptible if their MIC values were less than or equal to this clinical parameter.

### 4.4. In Vivo Effect of Tildipirosin on Bacterial Colonization of Tonsils

The ability of tildipirosin to inhibit tonsillar colonization by *A. pleuropneumoniae*, *P. multocida* and *G. parasuis* was an evaluation in conventional pigs raised on a farm with endemic circulation of these 3 pathogens. The farm was in the northern region of the state of Rio Grande do Sul (Brazil) and had a total of 1800 sows in production. The farm was part of an integration system (~280,000 sows) and produced weaned piglets (~6 kg, 24 days old). A total of 153 piglets born from 10 commercial-hybrid Large white × Landrace parity 3 sows were included in this study. As illustrated in Figure 6, the 10 sows and their respective piglets were assigned to two groups named G1 and G2. Up to 24 h after birth, all piglets were identified with a numerical ear tattoo and, at that moment, all sows belonging to group G1 (*n* = 5 sows) received a single intramuscular (shank region) injection of tildipirosin (4 mg/kg, Zuprevo, MSD Brazil). All piglets born to these sows (*n* = 75) also received tildipirosin (injection in the neck region) on days 1, 7 and 14 of life. In parallel, animals belonging to the group G2 (5 sows and 78 piglets) were inoculated with PBS. At the days 1, 7, 14 and 21 a total of 20 piglets per group (4 piglets per sow, randomly selected) were physically restrained, and the tonsils were scraped with a plastic spoon (single use). The mucus was collected from the spoon with a sterile cotton swab and transferred to a microtube containing 500 μL of PBS. Samples were transported under refrigeration to the laboratory and processed immediately. During the study, cross-fostering of the piglets was not allowed.

### 4.5. Molecular Detection of Tonsillar Bacteria

The genomic DNA of the tonsillar samples was extracted automatically in the IndiMag 48S equipment (Indical Bioscience, Leipzig, Germany) using the IndiMag Pathogen Kit (Indical Bioscience, Leipzig, Germany) following the manufacture’s instruction. From the eluted DNA, Real-time PCRs were performed to detect *Actinobacillus pleuropneumoniae*, *Glaesserella parasuis* and *Pasteurella multocida*, according to previously described protocols [46,47,48]. Genomic DNA from *A. pleuropneumoniae* SV5 (K17 strain), *G. parasuis* SV5 (Nagasaki strain) and *P. multocida* A (field isolate, #11246) were used as positive controls in qPCRs. Thermocycling was performed in the QuantiStudio^TM^ 6 Flex Real-Time PCR System (TermoFisher, Waltham, MA, USA). The load of each bacterial species was determined by quantitative (q) PCR and expressed as cycle threshold (Ct), i.e., the number of PCR cycles counted until transition to the exponential growth of PCR products.

### 4.6. Statistical Analyses

The χ^2^ or Fisher’s exact tests were used for testing of hypotheses regarding the association of the *in vitro* susceptibility of *A. pleuropneumoniae* or *P. multocida* isolates to tildipirosin with other qualitative parameters related to the bacterial isolation, such as Central or Southern Brazilian states, pig age range (for *P. multocida*), bacterial isolation year, and the place where the necropsy was performed (farm or slaughterhouse), serovars (in the case of *A. pleuropneumoniae)* and, finally, the presence or absence of *pfhA* gene (in the case of *P. multocida*); data were expressed as percentage of isolates that were susceptible to tildipirosin.

Comparison of the *G. parasuis* load between samples collected from animals treated and not treated with tildipirosin was performed using Two-Way ANOVA, and the difference between groups was evaluated with the Sindák multiple comparisons test. The normality assumption for the quantitative variable was assessed by the Kolmogorov-Smirnov test, and equality of variances was also assessed using Levene’s test. SPSS software version 26 (SPSS Inc., Armonk, NY, USA) and Prism 9 version 9.2.0 (GraphPad Software, LLC, San Diego, CA, USA) were used to conduct the statistical analyses. Statistical differences were considered when the *p*-values were < 0.05.

## 5. Conclusions

Tildipirosin displayed high *in vitro* activity against Brazilian clinical isolates of *A. pleuropneumoniae*, making it a suitable choice for the treatment of swine pleuropneumonia. Tildipirosin also exhibited substantial *in vitro* activity against *P. multocida* serogroup A, despite higher rates of non-susceptible isolates were observed for this microorganism. Based on our data, the use of tildipirosin can be strongly recommended for eradication programs targeting *A. pleuropneumoniae* and *P. multocida,* as well as for reducing tonsil colonization by *G. parasuis*. However, it is essential to conduct MIC testing procedures as a prerequisite for making informed antimicrobial treatment decisions and to monitor the evolution of *A. pleuropneumoniae* and *P. multocida* resistance to this modern macrolide under field conditions.

## Figures and Tables

**Figure 1 antibiotics-12-01658-f001:**
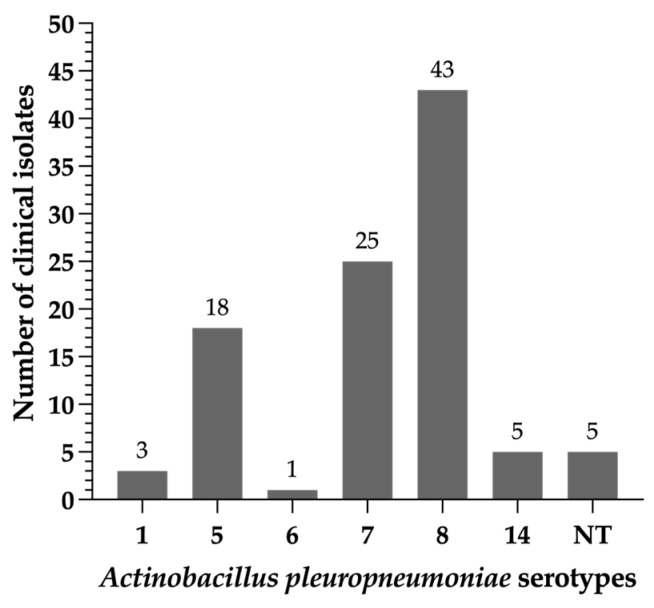
Distribution of serovars amongst the 100 *Actinobacillus pleuropneumoniae* clinical isolates recovered in Brazil between 2014 and 2018. The *A. pleuropneumoniae* isolates were typed by a multiplex capsular PCR typing system. NT: non-typeable strains.

**Figure 2 antibiotics-12-01658-f002:**
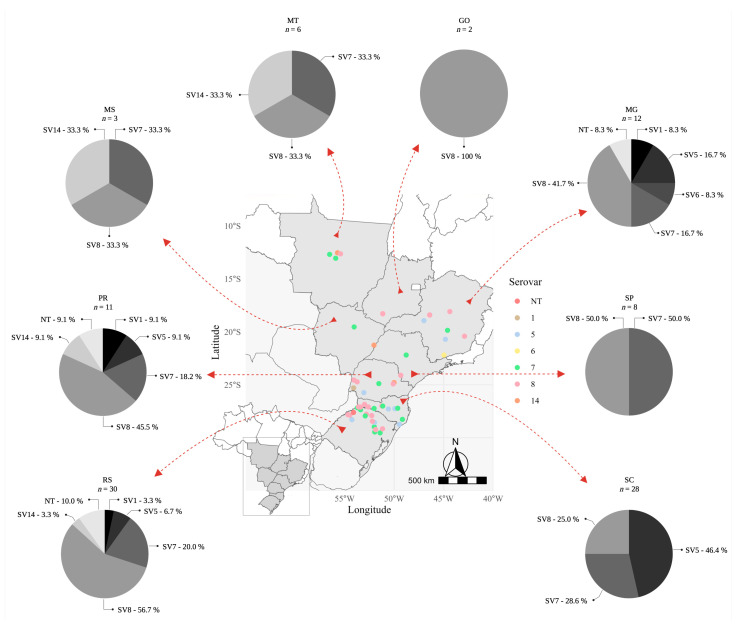
Geographic location of the different serovars of *Actinobacillus pleuropneumoniae* found in Brazil. The geographic location of the different serovars (colored dots) of *A. pleuropneumoniae* isolated in Brazil (from clinical cases or slaughterhouses) is illustrated in the central map. The serovar frequency isolation by state is indicated in the peripheral graphs (“parts of whole”). NT: non-typeable strains; SC: Santa Catarina; RS: Rio Grande do Sul; PR: Paraná; SP: São Paulo; MS: Mato Grosso do Sul; MT: Mato Grosso; MG: Minas Gerais; GO: Goiás.

**Figure 3 antibiotics-12-01658-f003:**
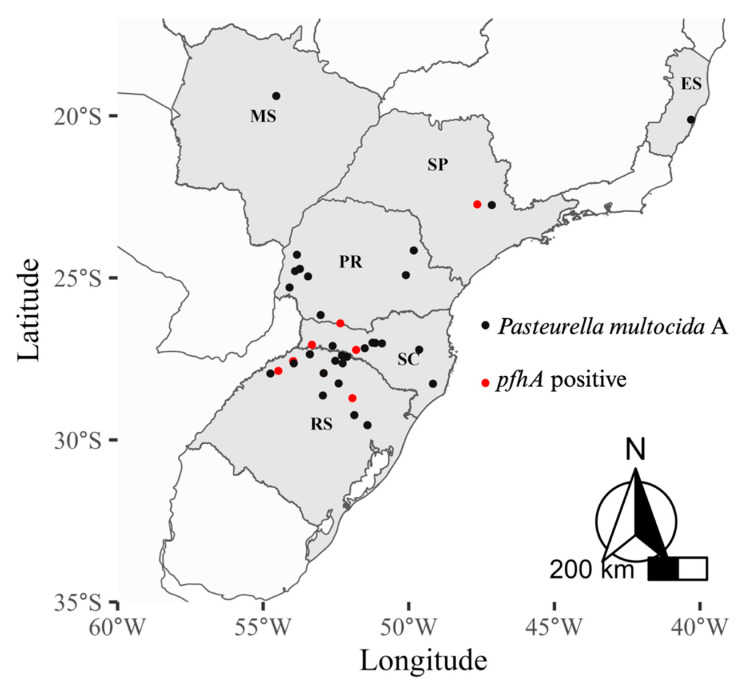
Geographic illustration of *Pasteurella multocida* serogroup A strains isolated in Brazil. The red dots indicate the location of the clinical strains that express the *pfhA* gene, which are considered of high pathogenicity. ES: Espírito Santo; MS: Mato Grosso do Sul; SP: São Paulo; PR: Paraná; SC: Santa Catarina; RS: Rio Grande do Sul.

**Figure 4 antibiotics-12-01658-f004:**
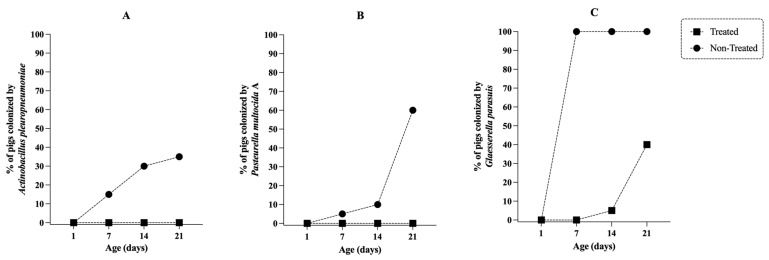
*Actinobacillus pleuropneumoniae*, *Pasteurella multocida* and *Glaesserella parasuis* colonization affected by tildipirosin treatment. During the lactation phase, tonsil scraping samples were collected from 20 piglets treated with tildipirosin (4 mg/kg, Intramuscularly) and born to treated sows (single dose after farrowing), and from 20 untreated piglets born to untreated dams. Samples were collected on days 1, 7, 14 and 21. Tildipirosin was applied to piglets on days 1, 7 and 14. *A. pleuropneumoniae*, *P. multocida* and *G. parasuis* DNA were tested by qPCR and the results are illustrated in graphics A, B and C, respectively.

**Figure 5 antibiotics-12-01658-f005:**
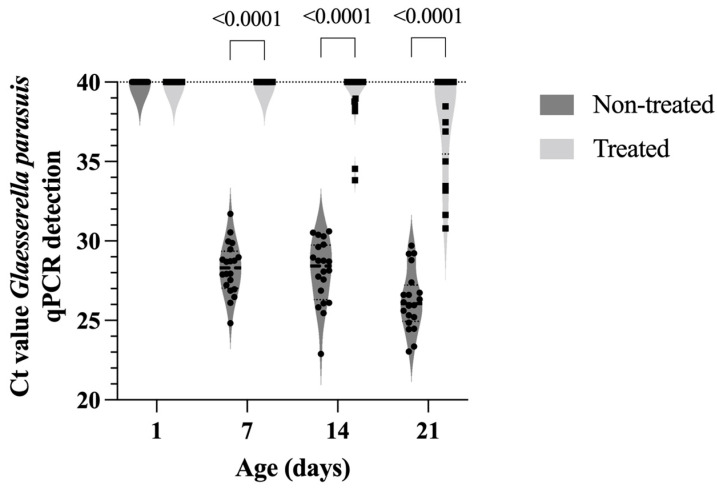
Molecular load of *Glaesserella parasuis* in tonsils of piglets treated and not treated with tildipirosin. The determination of the load of *G. parasuis* was determined by qPCR and the results are expressed as cycle threshold (Ct) of each sample. Samples (tonsil scrapings) were collected on days 1, 7, 14 and 21. Statistical differences between the two groups are indicated in the figure.

**Figure 6 antibiotics-12-01658-f006:**
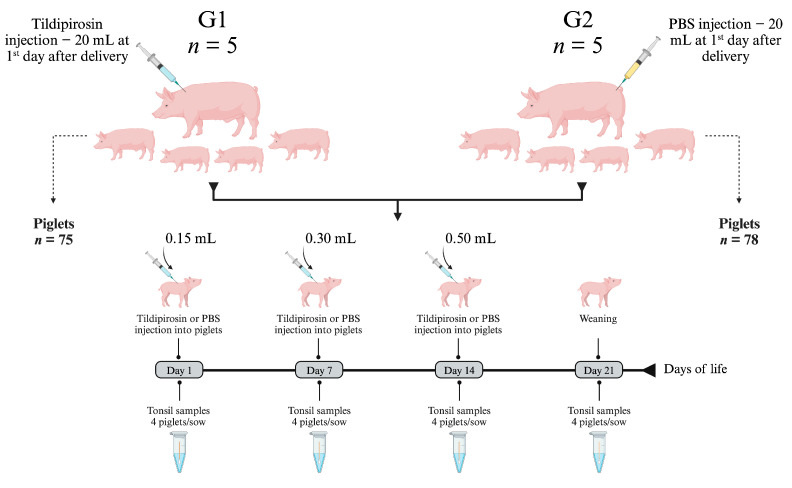
Experimental design for the clinical evaluation of tildipirosin. The antibiotic tildipirosin (Zuprevo^®^ 4%, MSD) and PBS were applied intramuscularly, and a new needle was used for each application. To calculate the therapeutic dose of the product (4 mg/kg) that would be applied, the piglets were weighed, and the average was used as a parameter for the calculation. For the sows, a weight of 210 kg was used for the calculation (estimated by the veterinarian responsible for the farm). Figure created with Biorender.com (accessed on 29 July 2023).

**Table 1 antibiotics-12-01658-t001:** MIC range, MIC_50_, MIC_90_ and percentage of susceptibility to tildipirosin of *Actinobacillus pleuropneumoniae* or *Pasteurella multocida* subsp. *multocida* isolates recovered in Brazil.

Tildipirosin Concentration and Clinical Parameter	Number and Accumulative% of Strains with MIC Equal to or Less than the Indicated Concentration	
	App Isolates (*n* = 100)	PmA Isolates (*n* = 60)
≤0.06 μg/mL	1 (1.0%)	-
0.12 μg/mL	17 (18.0%)	-
0.25 μg/mL	32 (50.0%)	-
0.5 μg/mL	27 (77.0%)	9 (15.0%)
1 μg/mL	8 (85.0%)	28 (61.7%)
2 μg/mL	4 (89.0%)	4 (68.3%)
4 μg/mL	2 (91.0%)	3 (73.3%)
8 μg/mL	3 (94.0%)	-
16 μg/mL	1 (95.0%)	-
32 μg/mL	1 (96.0%)	1 (75.0%)
64 μg/mL	-	-
>64 μg/mL	4 (100.0%)	15 (100.0%)
MIC range (μg/mL)	≤0.06–>64	0.5–>64
MIC_50_ (μg/mL)	0.25	1
MIC_90_ (μg/mL)	4	≥64
Clinical breakpoint (μg/mL)	≤16 ^$^	≤4 ^$^
Susceptibility rate	95.0	73.3

^$^ Susceptible clinical breakpoint according to CLSI, 2018.

## Data Availability

Data are contained within the article and Appendix A.

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
