# Peer review of "Brazilian Clinical Strains of Actinobacillus pleuropneumoniae and Pasteurella multocida: Capsular Diversity, Antimicrobial Susceptibility (In Vitro) and Proof of Concept for Prevention of Natural Colonization by Multi-Doses Protocol of Tildipirosin"

_antibiotics, 2023, doi:10.3390/antibiotics12121658_

Round 1

Reviewer 1 Report

Comments and Suggestions for Authors

The manuscript of Kuchiishi et al. describes the characterization of pig pneumonial pathogens from Brazil. The study design and implementation is appropriate as the standards of communication. Two major pathogens (A. pleuropneumoniae and P. multicoda) were cultured and the possible application of tildispirin against colonization was examined.

Minor comment

Line 19: expound and use your abbreviations Pm or PmA consistently

Reviewer 2 Report

Comments and Suggestions for Authors

Major issues

-The objectives of the study must be clearly defined and described in detail. As is now the relevant passage, it is not well-written and readers cannot get a clear picture of the content of the manuscript.

-4.1. and 4.2. Please describe the criteria used for the selection of the bacterial isolated tested in this study.

-4.6. Did you carry out a multivariable analysis to test dominant variables for the outcomes assessed? A model can be set up for multivariable analysis and appropriate evaluation will reveal possible interactions between the various variables of interest. Also, did you carry out collinearity assessment for the variables of interest?

-2.5. Please add a table with the results of the in vivo study in the revised manuscript.

-Please add a paragraph in Discussion to address concerns regarding development of antibiotic resistance as the consequence of applying the practice evaluated in clinical work.

Minor issues.

-Tildipirosin is a little-used and relatively-new antibiotic, hence please add a new paragraph in the Introduction to present the drug.

-Table 1 is badly formatted.

-Discussion can be separated in three sub-sections: A. pleuropneumoniae-related findings, P. multocida-related findings and combination and holistic discussion.

Overall.

A good study that can be published after significant revision as indicated.

Round 2

Reviewer 2 Report

Comments and Suggestions for Authors

Can the authors please try a multivariable model for their findings, as suggested in the initial review? This will greatly improve the final manuscript.

Author Response

Dear reviewer,

Thanks again for the review. We would like to inform you that we carried out the suggested statistical analysis. I also share with you that we sent the article for a statistical review, in this case, carried out by Dr. Arlei Coldebella, from Embrapa Swine and Poultry - Brazil, who has consolidated experience in the area. Therefore, the data and statistical methodology of this manuscript were reviewed by a statistics specialist.

After a detailed analysis of the database, with a reduced sample number, despite the representativeness of the set of isolates and variable responses, a logistical analysis was chosen (annexes).

In this regard, we find:

Concerning A pleuropeumoniae, the biggest problem is that the number of resistant samples is very low (n = 5) and therefore the logistic regression analysis will not be adequate due to the almost complete separation of factor levels. Anyway, we ran an analysis with all the factors that were in the table. We observed that nothing was significant, disregarding the problem of almost separation.

In the case of P multocida, although there are more resistant samples, there is also the problem of almost separating the data when analyzing the complete model.

In both cases, we also ran a logistic regression analysis with a stepwise method for variable selection. In both cases, the modeling did not identify any factor that should remain in the final model.

Therefore, we understand that the best path is to maintain the analysis initially presented. Due to the small number of resistant samples, Fisher's Exact test is the most recommended for this type of analysis. Furthermore, the test of each factor individually did not show a significant effect of the factors independently, which in itself indicates that the evaluation of more complex and complete models would probably not identify factors with significant effects on the response variable. Additionally, due to the low number of resistant samples, a logistic regression analysis model evaluating the various factors in the modeling faces the problem of almost separating the data, making it difficult to carry out the analysis.

Finally, when reanalyzing the data, we found an error regarding A. pleuropneumoniae serotype 1. The susceptibility rate for serotype 1 is 100% and not zero, as stated in version R1. This bug was fixed in version R2.

Thank you again for your excellent review and we remain at your disposal.

We hope that the article can be accepted for publication in Antibiotics.

Best regards,

Rafael
